# Pricing Compound and Extendible Options under Mixed Fractional Brownian Motion with Jumps

**Foad Shokrollahi**

Department of Mathematics and Statistics, University of Vaasa, P.O. Box 700, FIN-65101 Vaasa, Finland; foad.shokrollahi@uva.fi

**Abstract:** This study deals with the problem of pricing compound options when the underlying asset follows a mixed fractional Brownian motion with jumps. An analytic formula for compound options is derived under the risk neutral measure. Then, these results are applied to value extendible options. Moreover, some special cases of the formula are discussed, and numerical results are provided.

**Keywords:** pricing; mixed fractional Brownian motion; extendible and compound options; poisson process

---

## 1. Introduction

Compound option is a standard option with mother standard option being the underlying asset. Compound options have been extensively used in corporate finance. When the total value of a firm's assets is regarded as the risky underlying asset, the various corporate securities can be valued as claim contingent on underlying asset, and the option on the security is termed a compound option. The compound option models were first used by Geske [1] to value options on a share of common stock. Richard [2] extended Geske's work and obtained a closed-form solution for the price of an American call. Selby and Hodges [3] studied the valuation of compound options.

Extendible options are a generalized form of compound options whose maturities can be extended on the maturity date, at the choice of the option holder, and this extension may require the payment of an additional premium. They are widely applied in financial fields such as real estate, junk bonds, warrants with exercise price changes, and shared-equity mortgages, so many researchers carry out the theoretical models for pricing the options.

Prior valuation of extendible bonds was presented by Brennan et al. [4] and Ananthanaray et al. [5]. Longstal [6] extended their work to develop a set of pricing model for a wide variety of extendible options. Since these models assume the asset price follows geometric Brownian motion, they are unlikely to translate the abnormal vibrations in asset price when the arrival of important new information come out. Merton [7] considered the impact of a sudden event on the asset price in the financial market and proposed a geometric Brownian motion with jumps to match the abnormal fluctuation of financial asset price, which was introduced into derivation of the option pricing model. Based on this theory, Dias and Rocha [8] considered the problem of pricing extendible options under petroleum concessions in the presence of jumps. Kou [9] and Cont and Tankov [10] also considered the problem of pricing options under a jump diffusion environment in a larger setting. Moreover, Gukhal [11] derived a pricing model for extendible options when the asset dynamics were driven by jump diffusion process. Hence, the analysis of compound and extendible options by applying jump process is a significant issue and provides the motivation for this paper.

All this research above assumes that the logarithmic returns of the exchange rate are independent identically distributed normal random variables. However, the empirical studies demonstrated that the distributions of the logarithmic returns in the asset market generally reveal excess kurtosis with

more probability mass around the origin and in the tails and less in the flanks than what would occur for normally distributed data [10]. It can be said that the properties of financial return series are nonnormal, nonindependent, and nonlinear, self-similar, with heavy tails, in both autocorrelations and cross-correlations, and volatility clustering [12–16]. Since fractional Brownian motion (*FBM*) has two substantial features such as self-similarity and long-range dependence, thus using it is more applicable to capture behavior from financial asset [17–21]. Unfortunately, due to *FBM* is neither a Markov process nor a semimartingale, we are unable to apply the classical stochastic calculus to analyze it [22]. To get around this problem and to take into account the long memory property, it is reasonable to use the mixed fractional Brownian motion (*MFBM*) to capture fluctuations of the financial asset [23,24]. The *MFBM* is a linear combination of Brownian motion and *FBM* processes. Cheridito [23] proved that, for $H \in (3/4, 1)$, the mixed model with dependent Brownian motion and *FBM* was equivalent to one with Brownian motion, and hence it is arbitrage-free. For $H \in (\frac{1}{2}, 1)$, Mishura and Valkeila [25] proved that the mixed model is arbitrage-free.

In this paper, to capture the long-range property, to exclude the arbitrage in the environment of *FBM* and to get the jump or discontinuous component of asset prices, we consider the problem of compound option in a jump mixed fractional Brownian motion (*JMFBM*) environment. We then exert the result to value extendible options. We also provide representative numerical results. The *JMFBM* is based on the assumption that the underlying asset price is generated by a two-part stochastic process: (1) small, continuous price movements are generated by a *MFBM* process, and (2) large, infrequent price jumps are generated by a Poisson process. This two-part process is intuitively appealing, as it is consistent with an efficient market in which major information arrives infrequently and randomly. The rest of this paper is as follows. In Section 2, we briefly state some definitions related to *MFBM* that will be used in forthcoming sections. In Section 3, we analyze the problem of pricing compound option whose values follow a *JMFBM* process and present an explicit pricing formula for compound options. In Section 4, we derive an analytical valuation formula for pricing extendible option by compound option approach with only one extendible maturity under risk neutral measure, then extend this result to the valuation of an option with $N$ extendible maturity. Section 5 deals with the simulation studies for our pricing formula. Moreover, the comparison of our *JMFBM* model and traditional models is undertaken in this section. Section 6 is assigned to conclusion.

## 2. Auxiliary Facts

In this section, we recall some definitions and results which we need for the rest of paper [21,24,25].

**Definition 1.** *A MFBM of parameters $\epsilon, \alpha$ and $H$ is a linear combination of FBM and Brownian motion, under probability space $(\Omega, F, P)$ for any $t \in R^+$ by:*

$$M_t^H = \epsilon B_t + \alpha B_t^H, \tag{1}$$

*where $B_t$ is a Brownian motion , $B_t^H$ is an independent FBM with Hurst parameter $H \in (0, 1)$, $\epsilon$ and $\alpha$ are two real invariant such that $(\epsilon, \alpha) \neq (0, 0)$.*

Consider a frictionless continuous time economy where information arrives both continuously and discontinuously. This is modeled as a continuous component and as a discontinuous component in the price process. Assume that the asset does not pay any dividends. The price process can hence be specified as a superposition of these two components and can be represented as follows:

$$\begin{aligned} dS_t &= S_t(\mu - \lambda\kappa)dt + \sigma S_t dB_t \\ &+ \sigma S_t dB_t^H + (J-1)S_t dN_t, \ 0 < t \leq T, \ S_{T_0} = S_0, \end{aligned} \tag{2}$$

where $\mu, \sigma, \lambda$ are constant, $B_t$ is a standard Brownian motion, $B_t^H$ is an independent *FBM* and with Hurst parameter $H$, $N_t$ is a Poisson process with rate $\lambda$, $J - 1$ is the proportional change due to the

jump and $k \sim N(\mu_J = \ln(1+k) - \frac{1}{2}\sigma_J^2, \sigma_J^2)$. The Brownian motion $B_t$, the *FBM*, $B_t^H$, the Poisson process $N_t$ and the jump amplitude $J$ are independent.

Using Ito's Lemma [26], the solution for stochastic differential Equation (2) is

$$S_t = S_0 \exp\left[(r - \lambda k)t + \sigma B_t + \sigma B_t^H - \frac{1}{2}\sigma^2 t - \frac{1}{2}\sigma^2 t^{2H}\right] J(N(t)). \tag{3}$$

where $J(n) = \prod_{i=1}^n J_i$ for $n \geq 1$, $J_t$ is independently and identically distributed and $J_0 = 1$; $n$ is the Poisson distributed with parameter $\lambda t$. Let $x_t = \ln\frac{S_t}{S_0}$. From Equation (3) easily get

$$dx_t = \left(r - \lambda k - \frac{1}{2}\sigma^2 - H\sigma^2 t^{2H-1}\right)dt + \sigma dB_t + \sigma dB_t^H + \ln(J)dN_t. \tag{4}$$

Consider a European call option with maturity $T$ and the strike price $K$ written on the stock whose price process evolves as in Equation (2). The value of this call option is known from [27] and is given by

$$
\begin{aligned}
&C(S_0, K, T - T_0) \\
&= \sum_{n=0}^{\infty} \frac{e^{-\lambda'(T-T_0)}(\lambda'(T-T_0))^n}{n!} S_0 \Phi(d_1) - K e^{r(T-T_0)}\Phi(d_2),
\end{aligned} \tag{5}
$$

where

$$
\begin{aligned}
d_1 &= \frac{\ln\frac{S_0}{K} + r_n(T-T_0) + \frac{1}{2}[\sigma^2(T-T_0) + \sigma^2(T^{2H} - T_0^{2H}) + n\sigma_J^2]}{\sqrt{\sigma^2(T-T_0) + \sigma^2(T^{2H} - T_0^{2H}) + n\sigma_J^2}}, \\
d_2 &= d_1 - \sqrt{\sigma^2(T-T_0) + \sigma^2(T^{2H} - T_0^{2H}) + n\sigma_J^2},
\end{aligned}
$$

where $\lambda' = \lambda(1+k)$, $r_n = r - \lambda k + \frac{n\ln(1+k)}{T-T_0}$ and $\Phi(.)$ is the cumulative normal distribution.

## 3. Compound Options

To derive a compound option pricing formula in a jump mixed fractional market, we make the following assumptions.

(i)　　　There are no transaction costs or taxes and all securities are perfectly divisible;
(ii)　　security trading is continuous;
(iii)　　there are no riskless arbitrage opportunities;
(iv)　　the short-term interest rate $r$ is known and constant through time;
(v)　　the underlying asset price $S_t$ is governed by the following stochastic differential equation

Consider a compound call option written on the European call $C(K, T_2)$ with expiration date $T_1$ and exercise price $K_1$, where $T_1 < T_2$. Assume $CC[C(K, T_2), K_1, T_1]$ denotes this compound option. This compound option is exercised at time $T_1$ when the value of the underlying asset, $C(S_1, K, T_1, T_2)$, exceeds the strike price $K_1$. When $C(S_1, K, T_1, T_2) < K_1$, it is not optimal to exercise the compound option and hence expires worthless. The asset price at which one is indifferent between exercising and not exercising is specified by the following relation:

$$C(S_1, K, T_1, T_2) = K_1. \tag{6}$$

Let, $S_1^*$ shows the price of indifference which can be obtained as the numerical solution of Equation (6). When it is optimal to exercise the compound option at time $T_1$, the option holder pays $K_1$ and receives the European call $C(K, T_1, T_2)$. This European call can in turn be exercised at time $T_2$ when $S_T$ exceeds $K$ and expires worthless otherwise. Hence, the cashflows to the compound option

are an outflow of $K_1$ at time $T_1$ when $S_1 > S_1^*$, a net cashflow at time $T_2$ of $S_T - K$ when $S_1 > S_1^*$ and $S_T > K$, and none in the other states. The value of the compound option is the expected present value of these cashflows as follows:

$$
\begin{aligned}
& CC\left[C(K, T_2), K_1, T_0, T_1\right] \\
= \ & E_{T_0}\left[e^{-r(T_2-T_0)}(S_T - K)\mathbf{1}_{S_T>K}\right] + E_{T_0}\left[e^{-r(T_1-T_0)}(-K_1)\mathbf{1}_{S_1>S_1^*}\right] \\
= \ & E_{T_0}\left[e^{-r(T_1-T_0)}E_{T_1}\left[e^{-r(T_2-T_1)}(S_T - K)\mathbf{1}_{S_T>K}\right]\mathbf{1}_{S_1>S_1^*}\right] \\
& - E_{T_0}\left[e^{-r(T_1-T_0)}K_1\mathbf{1}_{S_1>S_1^*}\right] \\
= \ & E_{T_0}\left[e^{-r(T_2-T_0)}C(S_1, K, T_1, T_2)\mathbf{1}_{S_1>S_1^*}\right] - E_{T_0}\left[e^{-r(T_1-T_0)}K_1\mathbf{1}_{S_1>S_1^*}\right]
\end{aligned}
\tag{7}
$$

where $C(S_1, K, T_1, T_2)$ is given in Equation (5).

The evaluation of the first and second expectation in Equation (7), can be complex due to the jumps in the asset price process. this can be conditioning the expectation on the number of jumps in the intervals $[T_0, T_1)$ and $[T_1, T_2]$ denoted by $n_1$ and $n_2$, respectively. Let $m = n_1 + n_2$ shows the total number of jumps in the interval $[T_0, T_2]$ and use the Poisson probabilities, we have

$$
\begin{aligned}
& E_{T_0}\left[e^{-r(T_2-T_0)}C(S_1, K, T_1, T_2)\mathbf{1}_{S_1>S_1^*}\right] \\
= \ & E_{T_0}\left[e^{-r(T_1-T_0)}E_{T_1}\left[e^{-r(T_2-T_1)}(S_T - K)\mathbf{1}_{S_T>K}\right]\mathbf{1}_{S_1>S_1^*}\right] \\
= \ & \sum_{n_1=0}^{\infty}\sum_{n_2=0}^{\infty}\frac{e^{-\lambda'(T_1-T_0)}(\lambda'(T_1-T_0))^{n_1}}{n_1!}\frac{e^{-\lambda'(T_2-T_1)}(\lambda'(T_2-T_1))^{n_2}}{n_2!} \\
& \times E_{T_0}\left[e^{-r(T_1-T_0)}E_{T_1}\left[e^{-r(T_2-T_1)}(S_T - K)\mathbf{1}_{S_T>K}\right]\mathbf{1}_{S_1>S_1^*}\big|n_1, n_2\right] \\
= \ & \sum_{n_1=0}^{\infty}\sum_{n_2=0}^{\infty}\frac{e^{-\lambda'(T_1-T_0)}(\lambda'(T_1-T_0))^{n_1}}{n_1!}\frac{e^{-\lambda'(T_2-T_1)}(\lambda'(T-T_1))^{n_2}}{n_2!} \\
& \times E_{T_0}\left[e^{-r(T_2-T_0)}(S_T - K)\mathbf{1}_{S_T>K}\mathbf{1}_{S_1>S_1^*}\big|n_1, n_2\right]
\end{aligned}
$$

The evaluation of this expectation requires the joint density of two Poisson weighted sums of correlated normal. From this point, we work with the logarithmic return, $x_t = \ln\frac{S_t}{S_0}$, rather than the stock price. It is important to know that the correlation between the logarithmic return $x_{T_1}$ and $x_{T_2}$ depend on the number of jumps in the intervals $[T_0, T_1)$ and $[T_1, T_2]$. Conditioning on the number of jumps $n_1$ and $n_2$, $x_{T_1}$ has a normal distribution with mean

$$
\begin{aligned}
\mu_{J_{T_1-T_0}} = \ & (r - \lambda k)(T_1 - T_0) - \frac{1}{2}\sigma^2(T_1 - T_0) \\
& - \frac{1}{2}\sigma^2(T_1^{2H} - T_0^{2H}) + n_1\left[\ln(1+k) - \frac{1}{2}\sigma_J^2\right] \\
\sigma_{J_{T_1-T_0}}^2 = \ & \sigma^2(T_1 - T_0) + \sigma^2(T_1^{2H} - T_0^{2H}) + n_1\sigma_J^2,
\end{aligned}
$$

and $x_{T_2} \sim N(\mu_{J_{T_2-T_0}}, \sigma_{J_{T_2-T_0}}^2)$ where

$$
\begin{aligned}
\mu_{J_{T_2-T_0}} = \ & (r - \lambda k)(T_2 - T_0) - \frac{1}{2}\sigma^2(T_2 - T_0) \\
& - \frac{1}{2}\sigma^2(T_2^{2H} - T_0^{2H}) + m\left[\ln(1+k) - \frac{1}{2}\sigma_J^2\right] \\
\sigma_{J_{T_2-T_0}}^2 = \ & \sigma^2(T_2 - T_0) + \sigma^2(T_2^{2H} - T_0^{2H}) + m\sigma_J^2.
\end{aligned}
$$

The correlation coefficient between $x_{T_2}$ and $x_{T_1}$ is as follows

$$\rho = \frac{cov(x_{T_1}, x_{T_2})}{\sqrt{var(x_{T_1}) \times var(x_{T_2})}}.$$

Evaluating the first expectation in Equation (7) gives

$$
\begin{aligned}
&E_{T_0}\left[e^{-r(T_2-T_0)}C(S_1, K, T_1, T)\mathbf{1}_{S_1 > S_1^*}\right]\\
&= \sum_{n_1=0}^{\infty}\sum_{n_2=0}^{\infty} \frac{e^{-\lambda'(T_1-T_0)}(\lambda'(T_1-T_0))^{n_1}}{n_1!}\frac{e^{-\lambda'(T_2-T_1)}(\lambda'(T_2-T_1))^{n_2}}{n_2!}\\
&\times \left[S_0\Phi_2(a_1, b_1, \rho) - Ke^{-r(T_2-T_0)}\Phi_2(a_2, b_2, \rho)\right]
\end{aligned}
\tag{8}
$$

where

$$
\begin{aligned}
a_1 &= \frac{\ln\frac{S_0}{S_1^*} + \mu_{J_{T_1-T_0}} + \sigma_{J_{T_1-T_0}}^2}{\sqrt{\sigma_{J_{T_1-T_0}}^2}}, \quad a_2 = a_1 - \sqrt{\sigma_{J_{T_1-T_0}}^2}\\
b_1 &= \frac{\ln\frac{S_0}{K} + \mu_{J_{T_2-T_0}} + \sigma_{J_{T_2-T_0}}^2}{\sqrt{\sigma_{J_{T_2-T_0}}^2}}, \quad b_2 = b_1 - \sqrt{\sigma_{J_{T_2-T_0}}^2}
\end{aligned}
$$

$\Phi(x)$ is the standard univariate cumulative normal distribution function and $\Phi_2(x, y, \rho)$ is the standard bivariate cumulative normal distribution function with correlation coefficient $\rho$.

The second expectation in Equation (7) can be evaluated to give

$$
\begin{aligned}
&E_{T_0}\left[e^{-r(T_1-T_0)}K_1\mathbf{1}_{S_1 > S_1^*}\right]\\
&= \sum_{n_1=0}^{\infty} \frac{e^{-\lambda'(T_1-T_0)}(\lambda'(T_1-T_0))^{n_1}}{n_1!}E_{T_0}\left[e^{-r(T_1-T_0)}K_1\mathbf{1}_{S_1 > S_1^*}|n_1\right]\\
&= \sum_{n_1=0}^{\infty} \frac{e^{-\lambda'(T_1-T_0)}(\lambda'(T_1-T_0))^{n_1}}{n_1!}K_1 e^{-r(T_1-T_0)}\Phi(a_2),
\end{aligned}
\tag{9}
$$

where $a_2$ is defined above. Then, the following result for a compound call option is obtained.

**Theorem 1.** *The value of a compound call option with maturity $T_1$ and strike price $K_1$ written on a call option, with maturity $T_2$, strike $K$, and whose underlying asset follows the process in Equation (2), is given by*

$$
\begin{aligned}
&CC\left[C(K, T_2), K_1, T_0, T_1\right]\\
&= \Bigg\{ \sum_{n_1=0}^{\infty}\sum_{n_2=0}^{\infty} \frac{e^{-\lambda'(T_1-T_0)}(\lambda'(T_1-T_0))^{n_1}}{n_1!}\frac{e^{-\lambda'(T_2-T_1)}(\lambda'(T_2-T_1))^{n_2}}{n_2!}\\
&\times \left[S_0\Phi_2(a_1, b_1, \rho) - Ke^{-r(T_2-T_0)}\Phi_2(a_2, b_2, \rho)\right]\Bigg\}\\
&- \sum_{n_1=0}^{\infty} \frac{e^{-\lambda'(T_1-T_0)}(\lambda'(T_1-T_0))^{n_1}}{n_1!}K_1 e^{-r(T_1-T_0)}\Phi(a_2)
\end{aligned}
$$

*where $a_1, a_2, b_1, b_2,$ and $\rho$ are as defined previously.*

For a compound option with dividend payment rate $q$, the result is similar with Theorem 2, only $r$ replaces with $r - q$.

## 4. Extendible Option Pricing Formulae

Based on the assumptions in the last Section, let *EC* be the value of an extendible call option with time to expiration of $T_1$. At the time to expiration $T_1$, the holder of the extendible call can

(1)　let the call expire worthless if $S_{T_1} < L$, or
(2)　exercise the call and get $S_{T_1} - K_1$ if $S_{T_1} > M$, or
(3)　make a payment of an additional premium $A$ to extend the call to $T_2$ with a new strike of $K_2$ if $L \leq S_{T_1} \leq M$,

where $S_{T_1}$ is the underlying asset price and strike price at time $T_1$ , $K_1$ is the strike price at time $T_1$, and Longstaff [6] refers to $L$ and $M$ as critical values, where $L < M$.

If at expiration time $T_1$ the option is worth more than the extendible value with a new strike price of $K_2$ for a fee of $A$ for extending the expiration time $T_1$ to $T_2$, then it is best to exercise; that is, $S_{T_1} - K_1 \geq C(S_{T_1}, K_2, T_2 - T_1) - A$. Otherwise, it is best to extend the expiration time of the option to $T_2$ and exercise when it is worth more than zero; that is, $C(S_{T_1}, K_2, T_2 - T_1) - A > 0$. Moreover, the holder of the option should be impartial between extending and not exercising at value $L$ and impartial between exercising and extending at value $M$. Therefore, the critical values $L$ and $M$ are unique solutions of $M - K_1 = C(M, K_2, T_2 - T_1) - A$ and $M - K_1 = C(L, K_2, T_2 - T_1) - A = 0$. See Longstaff [6] and Gukhal [11] for an analysis of the conditions.

The value of a call option, $C$ at time $T_1$ with a time to expiration extended to $T_2$, as the discounted conditional expected payoff is given by

$$
\begin{aligned}
EC(S_0, K_1, T_1, K_2, T_2, A) \ &= \ E_{T_0}\left[e^{-r(T_1-T_0)}(S_{T_1} - K_1)\mathbf{1}_{S_{T_1}>M}\right] \\
&+ \ E_{T_0}\left[e^{-r(T_1-T_0)}\left(C(S_{T_1}, K_2, T_2 - T_1) - A\right)\mathbf{1}_{L \leq S_{T_1} \leq M}\right] \\
&= \ E_{T_0}\left[e^{-r(T_1-T_0)}(S_{T_1} - K_1)\mathbf{1}_{S_{T_1}>M}\right] \\
&+ \ E_{T_0}\left[e^{-r(T_1-T_0)}\left(C(S_{T_1}, K_2, T_2 - T_1) - A\right)\right. \\
&\times \ \left.\left(\mathbf{1}_{S_{T_1} \geq L} - \mathbf{1}_{S_{T_1} \geq M}\right)\right].
\end{aligned}
\tag{10}
$$

Then, by the same way of the call compound option, we have

$$
\begin{aligned}
&E_{T_0}\left[e^{-r(T_1-T_0)}(S_{T_1} - K_1)\mathbf{1}_{S_{T_1}>M}\right] \\
&= \ \sum_{n_1=0}^{\infty} \frac{e^{-\lambda'(T_1-T_0)}(\lambda'(T_1 - T_0))^{n_1}}{n_1!} E_{T_0}\left[e^{-r(T_1-T_0)}(S_{T_1} - K_1)\mathbf{1}_{S_{T_1}>M}|n_1\right],
\end{aligned}
\tag{11}
$$

$$
E_{T_0}\left[e^{-r(T_1-T_0)}\left(C(S_{T_1},K_2,T_2-T_1)-A\right)\left(\mathbf{1}_{S_{T_1}\geq L}-\mathbf{1}_{S_{T_1}\geq M}\right)\right]
$$

$$
= E_{T_0}\left[e^{-r(T_1-T_0)}E_{T_1}\left(e^{-r(T_2-T_1)}(S_{T_2}-K_2)\mathbf{1}_{S_{T_2}>K_2}\right)\left(\mathbf{1}_{S_{T_1}\geq L}-\mathbf{1}_{S_{T_1}\geq M}\right)\right]
$$

$$
- E_{T_0}\left[e^{-r(T_1-T_0)}A\left(\mathbf{1}_{S_{T_1}\geq L}-\mathbf{1}_{S_{T_1}\geq M}\right)\right]
$$

$$
= \left\{\sum_{n_1=0}^{\infty}\sum_{n_2=0}^{\infty}\frac{e^{-\lambda'(T_1-T_0)}(\lambda'(T_1-T_0))^{n_1}}{n_1!}\frac{e^{-\lambda'(T_2-T_1)}(\lambda'(T_2-T_1))^{n_2}}{n_2!}\right.
$$

$$
\times \left. E_{T_0}\left[e^{-r(T_2-T_0)}(S_{T_2}-K_2)\mathbf{1}_{S_{T_2}>K_2}\mathbf{1}_{S_{T_1}>L}|n_1,n_2\right]\right\}
$$

$$
- \left\{\sum_{n_1=0}^{\infty}\sum_{n_2=0}^{\infty}\frac{e^{-\lambda'(T_1-T_0)}(\lambda'(T_1-T_0))^{n_1}}{n_1!}\frac{e^{-\lambda'(T_2-T_1)}(\lambda'(T_2-T_1))^{n_2}}{n_2!}\right.
$$

$$
\times \left. E_{T_0}\left[e^{-r(T_2-T_0)}(S_{T_2}-K_2)\mathbf{1}_{S_{T_2}>K_2}\mathbf{1}_{S_{T_1}>M}|n_1,n_2\right]\right\}
$$
(12)
$$
- \left\{\sum_{n_1=0}^{\infty}\frac{e^{-\lambda'(T_1-T_0)}(\lambda'(T_1-T_0))^{n_1}}{n_1!}E_{T_0}\left[e^{-r(T_1-T_0)}A(\mathbf{1}_{S_{T_1}>L}|n_1-\mathbf{1}_{S_{T_1}>M}|n_1)\right]\right\}.
$$

Now, we assume that the asset price satisfies in Equation (2). Then, by calculating the expectations in Equations (11) and (12), the following result is derived.

**Theorem 2.** *The price of an extendible call option with time to expiration $T_1$ and strike price $K_1$, whose expiration time can extend to $T_2$ with a new strike price $K_2$ by the payment of an additional premium $A$, is given by*

$$
EC(S_t,K_1,T_1,K_2,T_2,A)
$$

$$
= \sum_{n_1=0}^{\infty}\frac{e^{-\lambda'(T_1-T_0)}(\lambda'(T_1-T_0))^{n_1}}{n_1!}\left[S_0\Phi(a_1)-K_1e^{-r(T_1-T_0)}\Phi(a_2)\right]
$$

$$
+ \left\{\sum_{n_1=0}^{\infty}\sum_{n_2=0}^{\infty}\frac{e^{-\lambda'(T_1-T_0)}(\lambda'(T_1-T_0))^{n_1}}{n_1!}\frac{e^{-\lambda'(T_2-T_1)}(\lambda'(T_2-T_1))^{n_2}}{n_2!}\right.
$$

$$
\times\left. \left[S_0\Phi_2(b_1,c_1,\rho)-K_2e^{-r(T_2-T_0)}\Phi(b_2,c_2,\rho)\right]\right\}
$$

$$
- \left\{\sum_{n_1=0}^{\infty}\sum_{n_2=0}^{\infty}\frac{e^{-\lambda'(T_1-T_0)}(\lambda'(T_1-T_0))^{n_1}}{n_1!}\frac{e^{-\lambda'(T_2-T_1)}(\lambda'(T_2-T_1))^{n_2}}{n_2!}\right.
$$
(13)
$$
- \left. \left[S_0\Phi_2(a_1,c_1,\rho)-K_2e^{-r(T_2-T_0)}\Phi(a_2,c_2,\rho)\right]\right\}
$$

$$
- \left\{\sum_{n_1=0}^{\infty}\frac{e^{-\lambda'(T_1-T_0)}(\lambda'(T_1-T_0))^{n_1}}{n_1!}S_0Ae^{-r(T_1-T_0)}\right.
$$

$$
\times\left. \left[\Phi(b_2)-\Phi(a_2)\right]\right\},
$$

*where*

$$
a_1 = \frac{\ln\frac{S_0}{M}+\mu_{J_{T_1-T_0}}+\sigma_{J_{T_1-T_0}}^2}{\sqrt{\sigma_{J_{T_1-T_0}}^2}}, \quad a_2=a_1-\sqrt{\sigma_{J_{T_1-T_0}}^2}
$$

$$
b_1 = \frac{\ln\frac{S_0}{L}+\mu_{J_{T_1-T_0}}+\sigma_{J_{T_1-T_0}}^2}{\sqrt{\sigma_{J_{T_1-T_0}}^2}}, \quad b_2=b_1-\sqrt{\sigma_{J_{T_1-T_0}}^2}
$$

$$
c_1 = \frac{\ln\frac{S_0}{K_2}+\mu_{J_{T_2-T_0}}+\sigma_{J_{T_2-T_0}}^2}{\sqrt{\sigma_{J_{T_2-T_0}}^2}}, \quad c_2=c_1-\sqrt{\sigma_{J_{T_2-T_0}}^2}
$$

$\Phi(x)$ is the standard univariate cumulative normal distribution function and $\Phi_2(x,y,\rho)$ is the standard bivariate cumulative normal distribution function with correlation coefficient $\rho$.

**Corollary 1.** *If $H = \frac{1}{2}$, the asset price satisfies the Merton jump diffusion equation*

$$dS_t = S_t(\mu - \lambda\kappa)dt + \sigma S_t dB_t + (J-1)S_t dN_t, \ 0 < t \le T, \ S_{T_0} = S_0, \tag{14}$$

*then, our results are consistent with the findings in [11].*

When $\lambda = 0$, the asset price follows the *MFBM* model shown below

$$dS_t = S_t r dt + \sigma S_t dB_t + \sigma S_t dB_t^H. \tag{15}$$

and the formula (15) reduces to the diffusion case. The result is in the following.

**Corollary 2.** *The price of an extendible call option with time to expiration $T_1$ and strike price $K_1$, whose expiration time can extend to $T_2$ with a new strike price $K_2$ by the payment of an additional premium A and written on an asset following Equation (15) is*

$$
\begin{aligned}
&EC(S_t, K_1, T_1, K_2, T_2, A) \\
=\ & S_0\Phi(a_1) - K_1 e^{-r(T_1-T_0)}\Phi(a_2) \\
& + S_0\Phi_2(b_1, c_1, \rho) - K_2 e^{-r(T_2-T_0)}\Phi(b_2, c_2, \rho) \\
& - \left[ S_0\Phi_2(a_1, c_1, \rho) - K_2 e^{-r(T_2-T_0)}\Phi(a_2, c_2, \rho) \right] \\
& - A e^{-r(T_1-T_0)}\left[ \Phi(b_2) - \Phi(a_2) \right],
\end{aligned} \tag{16}
$$

*where*

$$a_1 = \frac{\ln\frac{S_0}{M} + r(T_1 - T_0) + \frac{\sigma^2}{2}(T_1 - T_0) + \frac{\sigma^2}{2}(T_1^{2H} - T_0^{2H})}{\sqrt{\sigma^2(T_1 - T_0) + \sigma^2(T_1^{2H} - T_0^{2H})}},$$

$$a_2 = a_1 - \sigma\sqrt{T_1^{2H} - T_0^{2H} + T_1 - T_0}$$

$$b_1 = \frac{\ln\frac{S_0}{L} + r(T_1 - T_0) + \frac{\sigma^2}{2}(T_1 - T_0) + \frac{\sigma^2}{2}(T_1^{2H} - T_0^{2H})}{\sqrt{\sigma^2(T_1 - T_0) + \sigma^2(T_1^{2H} - T_0^{2H})}},$$

$$b_2 = b_1 - \sigma\sqrt{T_1^{2H} - T_0^{2H} + T_1 - T_0}$$

$$c_1 = \frac{\ln\frac{S_0}{K_2} + r(T_2 - T_0) + \frac{\sigma^2}{2}(T_2 - T_0) + \frac{\sigma^2}{2}(T_2^{2H} - T_0^{2H})}{\sqrt{\sigma^2(T_2 - T_0) + \sigma^2(T_2^{2H} - T_0^{2H})}}.$$

$$c_2 = c_1 - \sigma\sqrt{T_2^{2H} - T_0^{2H} + T_2 - T_0}.$$

Let us consider an extendible option with $N$ extended maturity times, the result is presented in the following corollary.

**Corollary 3.** *The value of the extendible call expiring at time $T_1$, written on an asset whose price is governed by Equation (2) and whose maturity extend to $T_2 < T_3 <, ..., < T_{N+1}$ with new strike of $K_2, K_3, ..., K_{N+1}$ by the payment of corresponding premium of $A_1, A_2, ..., A_{N+1}$, is given by*

$$
\begin{aligned}
EC_N(S_0, K_1, T_0, T_1) \;=\; & \sum_{j=1}^{N+1} \left\{ \left[ S_0 \Phi_j(a_{1j}^*, R_j^*) - K_j e^{r(T_j - t)} \Phi(a_{2j}^*, R_j^*) \right] \right. \\
& \quad - \left[ S_0 \Phi_j(c_{1j}^*, R_j^*) - K_j e^{r(T_j - t)} \Phi(c_{2j}^*, R_j^*) \right] \\
& \quad \left. - A_j e^{r(T_j - t)} \left[ \Phi(b_{2j}^*, R_{-1j}^*) - \Phi(a_{2j}^*, R_{-1j}^*) \right] \right\}
\end{aligned}
\tag{17}
$$

*where* $A_0 = 0, \Phi_j(a_{1j}^*, R_j^*)$ *is the j-dimensional multivariate normal integral with upper limits of integration given by the j-dimensional vector* $a_{1j}^*$ *and correlation matrix* $R_j^*$ *and define* $a_{1j}^* = [a_1(M_1, T_1 - t), -a_1(M_2, T_2 - t), ..., -a_1(M_j, T_j - t)]$. *The same as* $\Phi_j(c_{1j}^*, R_j^*)$ *and* $\Phi_j(b_{2j}^*, R_j^*)$ *and define*

$$
\begin{aligned}
c_{1j}^* &= [b_1(L_1, T_1 - t), a_1(M_2, T_2 - t), ..., b_1(L_{j-1}, T_{j-1} - t), a_1(M_j, T_j - t)] \\
b_{2j}^* &= [b_2(L_1, T_1 - t), b_2(M_2, T_2 - t), ..., b_2(L_j, T_j - t)]
\end{aligned}
$$

*and* $\Phi_1(c_{1j}^*, R_j^*)$. $R_j^*$ *is a* $j \times j$ *diagonal matrix with correlated coefficient* $\rho_{p-1,p}$ *as the pth diagonal element, 0 and negative correlated coefficient* $\rho_{j-1,j}$, *respectively, as the first and the last diagonal element, and correlated coefficient* $\rho_{p-1,s}(s = p + 1, ..., j)$. *As to the rest of the elements, we note that* $\rho_{p-1,s}$ *is equal to negative correlated coefficient* $\rho_{pj}$ *when* $s = j$ *and* $\rho_{p-1,s}$ *is equal to zero when* $p = 1, s = 0, ..., p - 1$, *the term* $T_j$ *and* $M_j, L_j$ *respectively represents the jth "time instant" and the critical price as defined previously.*

As $N$ increases to infinity the exercise opportunities become continuous and hence the value of the approximate option will converge in the limit to the value of the extendible option. Thus, the values $EC_1, EC_2, EC_3, ...$ form a converging sequence and the limit of this sequence is the value of the extendible, i.e., $\lim_{N \to \infty} EC_N(S_0, K_1, T_0, T_1) = EC(S_0, K_1, T_0, T_1)$. To minimize the impact of this computational complexity, we use the Richardson extrapolation method [28] with two points. This technique uses the first two values of a sequence of a sequence to obtain the limit of the sequence and leads to the following equation,

$$
EC_2 = 2EC_1 - EC_0,
\tag{18}
$$

where $EC_2$ stands for the extrapolated limit using $EC_1$ and $EC_0$.

## 5. Numerical Studies

Table 1 provides numerical results for extendible call options when the underlying asset pays no dividends. Column (3) displays the value obtained using the Merton model and column (4) shows the results using the Gukhal [11] method. Column (5) indicates the results by the $JMFBM$ model and values using the Richardson extrapolation technique for $EC_1$ and $EC_0$ are shown in column (6). By comparing columns Merton, Gukhal, $JMFBM$ and Richardson in Table 1 for the low- and high-maturity cases, we conclude that the call option prices obtained by these valuation methods are close to each other.

Figure 1 shows the price of extendible call option difference by the Merton, Guukhal, and $JMFBM$ models, according to the primary exercise date $T_1$ and strike price $K_1$. Figure 2 plots the impact of jump intensity on the option values.

**Table 1.** Results by different pricing models. Here, $r = 0.1, \sigma = 0.1, L = 5, M = 15, A = 0.05, H = 0.8$, $S = 12, \sigma_J = 0.3, k = -0.004$.

| $T_1$ | $K$ | Merton | Gukhal | JMFBM | Richardson |
|---|---|---|---|---|---|
| 1 | 10 | 0.1127 | 0.11143 | 0.1228 | 0.1330 |
| 1 | 11 | 0.0960 | 0.0997 | 0.1075 | 0.1190 |
| 1 | 12 | 0.0812 | 0.0852 | 0.0922 | 0.1031 |
| 1 | 13 | 0.0687 | 0.0707 | 0.0768 | 0.0850 |
| 1 | 14 | 0.0587 | 0.0561 | 0.0615 | 0.0566 |
| 0.5 | 10 | 1.0347 | 0.7521 | 0.7799 | 0.5250 |
| 0.5 | 11 | 0.8387 | 0.6541 | 0.6783 | 0.5180 |
| 0.5 | 12 | 0.6662 | 0.5560 | 0.5768 | 0.4875 |
| 0.5 | 13 | 0.5412 | 0.4579 | 0.4753 | 0.4094 |
| 0.5 | 14 | 0.4598 | 0.3598 | 0.3738 | 0.2871 |

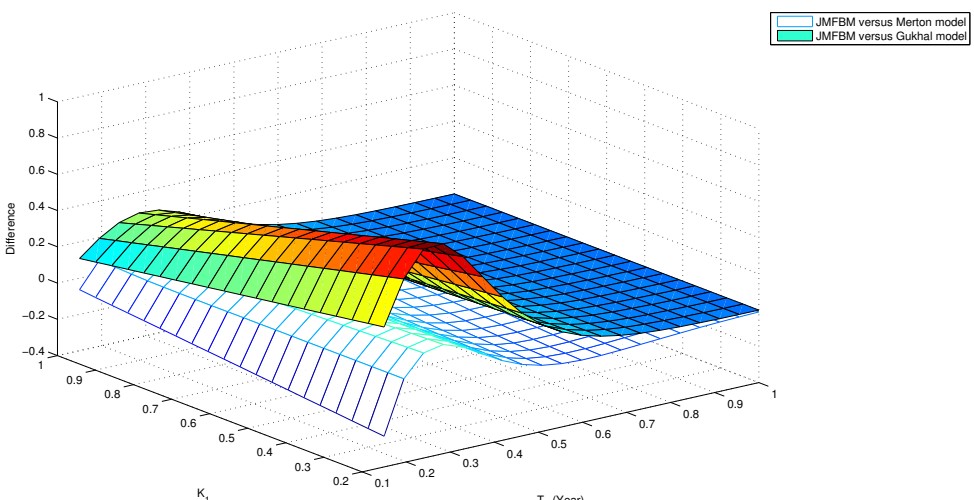

**Figure 1.** The relative difference between our *JMFBM*, Guukhal, and Merton models. Parameters fixed are $r = 0.3, \sigma = 0.4, L = 0.1, M = 1.5, A = 0.02, H = 0.8, S = 1.2, \sigma_J = 0.05, k = 0.4$ and $t = 0.1$.

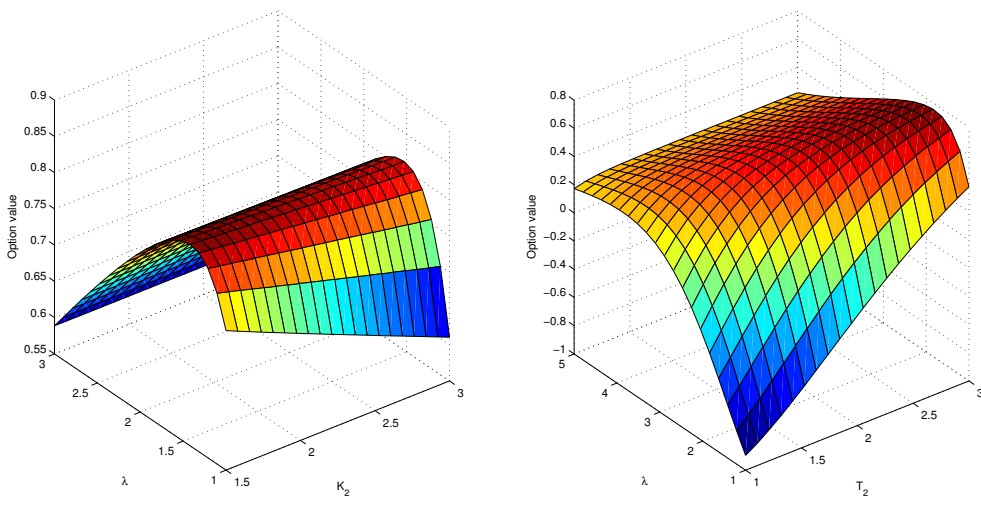

**Figure 2.** The impact of jump intensity on the option values. Parameters fixed are $r = 0.3, \sigma = 0.4$, $L = 0.1, M = 1.5, A = 0.02, H = 0.8, S = 1.2, \sigma_J = 0.05, k = 0.4$ and $t = 0.1$.

## 6. Conclusions

Mixed fractional Brownian motion is a strongly correlated stochastic process and jump is a significant component in financial markets. The combination of them provides better fit to evident observations because it can fully describe high frequency financial returns display, potential jumps, long memory, volatility clustering, skewness, and excess kurtosis. In this paper, we use a jump mixed fractional Brownian motion to capture the behavior of the underlying asset price dynamics and deduce the pricing formula for compound options. We then apply this result to the valuation of extendible options under a jump mixed fractional Brownian motion environment. Numerical results and some special cases are provided for extendible call options.

**Funding:** This research received no external funding.

**Acknowledgments:** I would like to thank referees and the editor for their careful reading and their valuable comments.

**Conflicts of Interest:** The author declares no conflict of interest.

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
