# Peer review of "Pricing Compound and Extendible Options under Mixed Fractional Brownian Motion with Jumps"

_axioms, doi:10.3390/axioms8020039_

Round 1
Reviewer 1 Report
The author extends previous work he has done applying a model with fractional Brownian motion (FBM) and jumps to vanilla options, applying his model here to compound options, so extending previous results from 30-40 years ago based on the Black-Scholes model. He cites two basic reasons for making the extension. The first is the observation that asset distributions in the market exhibit more kurtosis than is implied by the Black-Scholes model; consequently options tend to be under-priced away from the money. The second is that various stylised facts about the time series of financial returns are seen to be different from what is implied by adopting a Black-Scholes perspective.
The first of these issues is successfully addressed by the inclusion of jumps, as has been well-attested by numerous researchers. The second, however, does not specifically impact on option prices so it is not clear how attempts to address it would result in a better pricing model. Further, it is not brought out in the paper by comparison with evidence how either of the two features modelled improve on Geske’s formulae in [11], if at all. Further, although it is not brought out in the paper, it is demonstrated by the author in [25] that the only impact of the FBM on the terminal asset distribution is to modify the term structure of variance: rather than being a linear function of time, as results in the Black-Scholes model when a constant volatility is assumed, the dependence is superlinear when H > 0.5 is assumed. But this effect can be more simply achieved and with greater flexibility simply by allowing volatility to be a function of time, as has long been common practice. It is not therefore clear that anything has been achieved by the introduction of FBM to this problem.
Finally in the conclusion it is stated (without apparent justification) that the model “provides better fit to evident observations.” As stated above, this would appear to be generally true (albeit not demonstrated) for the jumps but not for the FBM. But the comparison being made here appears to be with the Black-Scholes model used by Geske in [11] 40 years ago. This is hardly a huge achievement, particularly as no evidence in relation to the specific problem of compound options is even alluded to.
The paper would in my opinion be improved by removing all mention of the FBM, simply allowing volatility to be a function of time and presenting formulae which show the impact of jumps alone on Geske’s results. This would I believe constitute a new and interesting result in itself. If no data can be found showing the smile-skew manifested by compound options (deviation of prices in/out of the money from Black-Scholes), data should be explicitly cited at least for vanilla options as this is readily available. Finally the author should look to demonstrate the degree of smile/skew resulting from the jump component of the model as a function of K and T. Specifically the impact of the jump intensity and assumed jump size distribution would be of interest.
Author Response
Dear Reviewer,
Thank you very much for your comments.
I have added pictures to show the impact of intensity jump size.
We dont have any information about FBM and just we have a little information about mixed fractional Brownian motion (MFBM).
Let me know if you have any comments.
Best Regards
Foad Shokrollahi

Reviewer 2 Report
see report

Author Response
Dear Reviewer,
Thank you very much for your comments.
I have done all of your comments.
Let me know if you have any comments.
Best Regards
Foad Shokrollahi
Round 2
Reviewer 1 Report
The author has addressed my main concern about bringing out the impact of the jump component which I believe is the main contribution of the paper. I still believe the author should look to demonstrate the level of smile (kurtosis) and skew resulting from the inclusion of the jump component by plotting not just option price but also the Black-Scholes implied vol, particularly as he has claimed this is the main shortcoming he seeks to address in the Black-Scholes model.
It would also be of interest to do the same exercise for the mixing parameter alpha to illustrate the degree to which skew/smile is induced by the inclusion of the FBM, but I suspect this might turn out not to be as important as the author implies.
Author Response
Dear Reviewer,
Your comment is good. But, find the skewness an kurtosis of our model and plot it are long story and it takes more time and it can be another paper.
Best Regards
Foad Shokrollahi
